# Predictors of Global Non-Motor Symptoms Burden Progression in Parkinson’s Disease. Results from the COPPADIS Cohort at 2-Year Follow-Up

**DOI:** 10.3390/jpm11070626

**Published:** 2021-06-30

**Authors:** Diego Santos-García, Teresa de Deus, Carlos Cores, Hector Canfield, Jose M Paz González, Cristina Martínez Miró, Lorena Valdés Aymerich, Ester Suárez, Silvia Jesús, Miquel Aguilar, Pau Pastor, Lluis Planellas, Marina Cosgaya, Juan García Caldentey, Nuria Caballol, Ines Legarda, Jorge Hernández-Vara, Iria Cabo, Lydia López Manzanares, Isabel González Aramburu, Maria A Ávila Rivera, Maria J Catalán, Victor Nogueira, Victor Puente, Julio Dotor, Carmen Borrué, Berta Solano, Maria Álvarez Sauco, Lydia Vela, Sonia Escalante, Esther Cubo, Francisco Carrillo, Juan C Martínez Castrillo, Pilar Sánchez Alonso, Gemma Alonso, Nuria López Ariztegui, Itziar Gastón, Jaime Kulisevsky, Marta Blázquez, Manuel Seijo, Javier Rúiz Martínez, Caridad Valero, Monica Kurtis, Oriol de Fábregues, Jessica Ardura, Ruben Alonso, Carlos Ordás, Luis M López Díaz, Darrian McAfee, Pablo Martinez-Martin, Pablo Mir

**Affiliations:** 1CHUAC, Complejo Hospitalario Universitario de A Coruña, 15006 A Coruña, Spain; Carlos.Cores.Bartolome@sergas.es (C.C.); jpazg1@hotmail.com (J.M.P.G.); Cristina.Martinez.Miro@sergas.es (C.M.M.); lorena.aymerich@gamil.com (L.V.A.); 2CHUF, Complejo Hospitalario Universitario de Ferrol, 15405 A Coruña, Spain; terecorreomovil@gmail.com (T.d.D.); canfield.hector@gmail.com (H.C.); str_sc@hotmail.com (E.S.); 3Unidad de Trastornos del Movimiento, Servicio de Neurología y Neurofisiología Clínica, Instituto de Biomedicina de Sevilla, Hospital Universitario Virgen del Rocío/CSIC/Universidad de Sevilla, 41009 Seville, Spain; smaestre-ibis@us.es (S.J.); pmir@us.es (P.M.); 4CIBERNED (Centro de Investigación Biomédica en Red Enfermedades Neurodegenerativas), 28031 Madrid, Spain; isagaramburu@gmail.com (I.G.A.); Jkulisevsky@santpau.cat (J.K.); pmm650@hotmail.com (P.M.-M.); 5Hospital Universitari Mutua de Terrassa, 08221 Barcelona, Spain; miquelaguilar@gmail.com (M.A.); paupastor@mutuaterrassa.es (P.P.); 6Hospital Clínic de Barcelona, 08036 Barcelona, Spain; lplanellas@hotmail.com (L.P.); marinacosgaya@gmail.com (M.C.); 7Centro Neurológico Oms 42, 07003 Palma de Mallorca, Spain; juangcaldentey@hotmail.com; 8Consorci Sanitari Integral, Hospital Moisés Broggi, 08970 Barcelona, Spain; nuriacaballol@hotmail.com; 9Hospital Universitario Son Espases, 07120 Palma de Mallorca, Spain; ines.legarda@ssib.es; 10Hospital Universitario Vall d’Hebron, 08035 Barcelona, Spain; hernandezvarajorge76@gmail.com (J.H.-V.); odefabregues@gmail.com (O.d.F.); 11Complejo Hospitalario Universitario de Pontevedra (CHOP), 36071 Pontevedra, Spain; icabol@yahoo.es (I.C.); manuel.seijo.martinez@sergas.es (M.S.); 12Hospital Universitario La Princesa, 28006 Madrid, Spain; lydialopez@hotmail.com; 13Hospital Universitario Marqués de Valdecilla, 39011 Santander, Spain; 14Consorci Sanitari Integral, Hospital General de L’Hospitalet, 08906 Barcelona, Spain; asuncion.avila@sanitatintegral.org; 15Hospital Universitario Clínico San Carlos, 28040 Madrid, Spain; mariajose.catalan@salud.madrid.org; 16Hospital Da Costa, 27880 Burela, Spain; victor.nogueira.fernandez@sergas.es; 17Hospital del Mar, 08003 Barcelona, Spain; Vpuente@parcdesalutmar.cat; 18Hospital Universitario Virgen Macarena, 41009 Sevilla, Spain; juliodotor@gmail.com; 19Hospital Infanta Sofía, 28702 Madrid, Spain; carmenborrue@hotmail.com; 20Institut d’Assistència Sanitària (IAS)—Institut Català de la Salut, 17190 Girona, Spain; berta_solano@hotmail.com; 21Hospital General Universitario de Elche, 03203 Elche, Spain; mariaalsa@hotmail.com; 22Fundación Hospital de Alcorcón, 28922 Madrid, Spain; lvela@fhalcorcon.es; 23Hospital de Tortosa Verge de la Cinta (HTVC), 43500 Tarragona, Spain; sescalant@yahoo.es; 24Complejo Asistencial Universitario de Burgos, 09006 Burgos, Spain; esthercubo@gmail.com; 25Hospital Universitario de Canarias, 38320 San Cristóbal de la Laguna, Spain; fcarpad@gobiernodecanarias.org; 26Hospital Universitario Ramón y Cajal, IRYCIS, 28034 Madrid, Spain; jcmcastrillo@gmail.com; 27Hospital Universitario Puerta de Hierro, 28222 Madrid, Spain; PISANCHEZAL@GMAIL.COM; 28Hospital Álvaro Cunqueiro, Complejo Hospitalario Universitario de Vigo (CHUVI), 36213 Vigo, Spain; gemavarita@gmail.com; 29Complejo Hospitalario de Toledo, 45071 Toledo, Spain; nlariztegui@gmail.com; 30Complejo Hospitalario de Navarra, 31008 Pamplona, Spain; itziar.gaston.zubimendi@cfnavarra.es; 31Hospital de Sant Pau, 08041 Barcelona, Spain; 32Hospital Universitario Central de Asturias, 33011 Oviedo, Spain; marta.blazquez.estrada@gmail.com; 33Hospital Universitario Donostia, 20009 San Sebastián, Spain; JAVIER.RUIZMARTINEZ@osakidetza.eus; 34Hospital Arnau de Vilanova, 46015 Valencia, Spain; carivalero@icloud.com; 35Hospital Ruber Internacional, 28034 Madrid, Spain; mkurtis@ruberinternacional.es; 36Hospital de Cabueñes, 33394 Gijón, Spain; jessardura@yahoo.es; 37Hospital Universitario Lucus Augusti (HULA), 27002 Lugo, Spain; r.alonso1408@gmail.com; 38Hospital Rey Juan Carlos, 28933 Madrid, Spain; carlos.ordas@quironsalud.es; 39Complejo Hospitalario Universitario de Orense (CHUO), 32005 Orense, Spain; Luis.Manuel.Lopez.Diaz@sergas.es; 40Laboratory for Cognition and Neural Stimulation, Univeristy of Pennsylvania, Philadelphia, PA 19104, USA; mcafeed@sas.upenn.edu

**Keywords:** mood, non-motor symptoms, Parkinson’s disease, progression, quality of life

## Abstract

**Background and Objective:** Non-motor symptoms (NMS) progress in different ways between Parkinson’s disease (PD) patients. The aim of the present study was to (1) analyze the change in global NMS burden in a PD cohort after a 2-year follow-up, (2) to compare the changes with a control group, and (3) to identify predictors of global NMS burden progression in the PD group. **Material and Methods:** PD patients and controls, recruited from 35 centers of Spain from the COPPADIS cohort from January 2016 to November 2017, were followed-up with after 2 years. The Non-Motor Symptoms Scale (NMSS) was administered at baseline (V0) and at 24 months ± 1 month (V2). Linear regression models were used for determining predictive factors of global NMS burden progression (NMSS total score change from V0 to V2 as dependent variable). **Results:** After the 2-year follow-up, the mean NMS burden (NMSS total score) significantly increased in PD patients by 18.8% (from 45.08 ± 37.62 to 53.55 ± 42.28; *p* < 0.0001; N = 501; 60.2% males, mean age 62.59 ± 8.91) compared to no change observed in controls (from 14.74 ± 18.72 to 14.65 ± 21.82; *p* = 0.428; N = 122; 49.5% males, mean age 60.99 ± 8.32) (*p* < 0.0001). NMSS total score at baseline (β = −0.52), change from V0 to V2 in PDSS (Parkinson’s Disease Sleep Scale) (β = −0.34), and change from V0 to V2 in NPI (Neuropsychiatric Inventory) (β = 0.25) provided the highest contributions to the model (adjusted R-squared 0.41; Durbin-Watson test = 1.865). **Conclusions:** Global NMS burden demonstrates short-term progression in PD patients but not in controls and identifies worsening sleep problems and neuropsychiatric symptoms as significant independent predictors of this NMS progression.

## 1. Introduction

Parkinson’s disease (PD), the second most common neurodegenerative disease after Alzheimer’s disease, is a progressive neurodegenerative disorder causing motor and non-motor symptoms (NMS) that result in disability, loss of patient autonomy, and caregiver burden [1]. Some NMS (e.g., olfactory disorders, constipation, or sleep disturbances) may precede motor symptoms and could be useful as prodromal/preclinical markers of PD [2]. Other symptoms, such as dementia and psychosis, more frequently develop during the late stages of the disease and are sometimes difficult to manage [3]. Early identification and proper management of NMS is important because NMS are common and negatively impact a patient’s quality of life (QoL) [4]. Many cross-sectional studies have analyzed the impact of NMS burden on QoL in PD patients [5,6,7,8]. However, there is a lack of knowledge about how NMS burden progresses over time and how its progression impacts on QoL [9,10,11,12]. Thus, there are limited data from large, controlled, prospective studies on the evolution of NMS in PD [12]. Very recently, we observed that NMS progression is an important factor which impacts the change in a patient’s QoL in PD patients from the COPPADIS cohort after a 2-year follow-up [13]. However, not all PD patients progress in the same way with regard to NMS. Identification of predictive factors for global NMS burden progression is necessary to be able to propose specific interventions (e.g., medication, other therapy, preventive action, etc.). In other words, avoiding or at least reducing the progression of NMS perceived by the patient could help avoid a deterioration in his/her QoL perception.

The aim of the present study was to (1) analyze the change in global NMS burden, and in some NMS in particular, in PD patients from the COPPADIS cohort after a 2-year follow-up, (2) to compare the changes with a control group, and (3) to identify predictors of global NMS burden progression in the PD group.

## 2. Methods

PD patients and controls, who were recruited from January 2016 to November 2017 from 35 centers of Spain from the COPPADIS cohort [14] and were evaluated again at 2-year follow-up, were included in the study. Methodology about COPPADIS-2015 study can be consulted at https://bmcneurol.biomedcentral.com/articles/10.1186/s12883-016-0548-9 (25 February 2016) [15]. This is a multi-center, observational, longitudinal-prospective, 5-year follow-up study designed to analyze disease progression in a Spanish population of PD patients. All patients included were diagnosed according to UK PD Brain Bank criteria. Exclusion criteria were: atypical parkinsonism, dementia (Mini Mental State Examination (MMSE) <26), age <18 or >75 years, inability to read or understand the questionnaires, to receive any advanced therapy (continuous infusion of levodopa or apomorphine, and/or with deep brain stimulation at baseline), and the presence of comorbidity, sequelae, or any disorder that could interfere with the assessment.

Information on sociodemographic aspects, factors related to PD, comorbidity, and treatment was collected. V0 (baseline visit) and V2 (2 years ± 1 month) evaluations included motor assessment (Hoenh & Yahr [H&Y], Unified Parkinson’s Disease Rating Scale (UPDRS) part III and part IV, Freezing of Gait Questionnaire (FOGQ)), non-motor symptoms (Non-Motor Symptoms Scale (NMSS), Parkinson’s Disease Sleep Scale (PDSS), Visual Analog Scale-Pain (VAS-Pain), Visual Analog Fatigue Scale (VAFS)), cognition (MMSE, Parkinson’s Disease Cognitive Rating Scale (PD-CRS), completing a simple 16-piece puzzle), mood and neuropsychiatric symptoms (Beck Depression Inventory-II (BDI-II), Neuropsychiatric Inventory (NPI), Questionnaire for Impulsive-Compulsive Disorders in Parkinson’s Disease-Rating Scale (QUIP-RS)), disability (Schwab & England Activities of Daily Living Scale (ADLS)), and QoL (the 39-item Parkinson’s disease Questionnaire (PDQ-39), PQ-10, the EUROHIS-QOL 8-item index (EUROHIS-QOL8)) [15]. In patients with motor fluctuations, the motor assessment was made during the OFF state (without medication in the last 12 h) and during the ON state. On the other hand, the assessment was only performed without medication in patients without motor fluctuations. The same evaluation as for the patients, except for the motor assessment, was performed in control subjects at V0 and at V2 (2 years ± 1 month). Furthermore, motor and non-motor assessment (NMSS and ADLS) was conducted in PD patients at 1 year ± 1 month [15].

The NMSS [16] was used for assessing the NMS and defining the global NMS burden at V0 and at V2 in both groups, PD patients and controls. It was applied in PD patients at V1 as well. The NMSS includes 30 items, each with a different non-motor symptom. The symptoms refer to the 4 weeks prior to assessment. The total score for each item is the result of multiplying the frequency (0, never; 1, rarely; 2, often; 3, frequent; 4, very often) x severity (1, mild; 2, moderate; 3, severe) and will vary from 0 to 12 points. The scale score ranges from 0 to 360 points (NMSS total score). The items are grouped into nine different domains: (1) Cardiovascular (items 1 and 2; score, 0 to 24); (2) Sleep/fatigue (items 3, 4, 5, and 6; score, 0 to 48); (3) Depression/apathy (items 7, 8, 9, 10, 11, and 12; score, 0 to 72); (4) Perceptual problems/hallucinations (items 13, 14, and 15; score, 0 to 36); (5) Attention/memory (items 16, 17, and 18; score, 0 to 36); (6) Gastrointestinal tract (items 19, 20, and 21; score 0 to 36); (7) Urinary symptoms (items 22, 23, and 24; score, 0 to 36); (8) Sexual dysfunction (items 25 and 26; score 0 to 24); and (9) Miscellaneous (items 27, 28, 29, and 30; score, 0 to 48). Each domain of the NMSS was expressed as a percentage: (score/total score) × 100. The global NMS burden was defined as the NMS total score and it was categorized in four groups: mild (NMSS 1–20); moderate (NMSS 21–40); severe (NMSS 41–70); very severe (NMSS >70) [17].

### 2.1. Data Analysis

Data were processed using SPSS 20.0 (IBM, Armonk, NY, USA) for Windows. For comparisons between groups, the Student’s *t*-test, Mann–Whitney U test, Chi-square test, or Fisher test was used as appropriate (distribution for variables was verified by one-sample Kolmogorov–Smirnov test).

General linear model (GLM) repeated measure was used to test whether the mean differences of the NMSS total score and the NMSS domain scores between the two visits (V0 and V2) were significant in both PD patients and controls. In PD patients, it was analyzed the change between V1 and V0, V2 and V1 and between all groups as well. Age, gender and LEDD (levodopa equivalent daily dose) at V0 and at V2 were included as covariates. This test was also applied for testing the difference between V2 and V0 in other variables. Cohen’s d formula was applied for measuring the effect size (in PD patients). It was considered: small effect = 0.2; medium effect = 0.5; large effect = 0.8. The Bonferroni method was used was used as a post-hoc test after ANOVA. Spearman’s or Pearson’s correlation coefficient, as appropriate, was used for analyzing the relationship between the change from V0 to V2 in continuous variables. Correlations were considered weak for coefficient values ≤0.29, moderate for values between 0.30 and 0.59, and strong for values ≥0.60. The *p*-value was considered significant when it was <0.05.

Linear regression models were used for determining predictive factors of global NMS burden progression (NMSS total score change from V0 to V2 as dependent variable). Any variable with univariate associations with *p*-values <0.20 were included in a multivariable model, and a backwards selection process was used to remove variables individually until all remaining variables were significant at the 0.10 level [12]. The variables considered for the analysis were: (1) at baseline (V0): age, gender, disease duration, total number of non-antiparkinsonian drugs, LEDD, daily dose of levodopa, daily dose of dopamine agonist, motor phenotype [18], UPDRS-III (OFF), UPDRS-IV, FOGQ, PD-CRS, NMSS, PDSS, NPI, QUIP-RS, VAS-PAIN, VASF-Physical, and VASF-Mental; (2) at 2-year follow-up (V2): to be receiving L-dopa, a MAO-B (monoamine oxidase type B) inhibitor, a COMT (catechol-o-methyl transferase) inhibitor, a dopamine agonist, and to practice regular exercise; (3) at the end of the follow-up (V2): change from V0 to V2 in total number of non-antiparkinsonian drugs, LEDD, daily dose of levodopa, daily dose of dopamine agonist, UPDRS-III, UPDRS-IV, FOGQ, PD-CRS, PDSS, NPI, QUIP-RS, VAS-PAIN, VASF-Physical, and VASF-Mental. Tolerance and variance inflation factor (VIF) were used to detect multicollinearity (multicollinearity was considered problematic when tolerance was less than 0.2 and, simultaneously, the value of VIF 10 and above). Finally, according to the change in the NMSS total score, patients were classified as “NMS—improver” (NMSS total score at V2 < NMSS total score at V0) or “non NMS improver” (NMSS total score at V2 ≥ NMSS total score at V0) and predictors of NMS improvement were identified. A level of *p* = 0.01 was considered for determining significant variables in the model.

### 2.2. Standard Protocol Approvals, Registrations, and Patient Consents

For this study, we received approval from the Comité de Ética de la Investigación Clínica de Galicia from Spain (2014/534; 2/12/2014). Written informed consent from all participants in this study were obtained before the start of the study. COPPADIS-2015 was classified by the AEMPS (Agencia Española del Medicamento y Productos Sanitarios) as a Post-authorization Prospective Follow-up study with the code COH-PAK-2014-01.

## 3. Results

After the 2-year follow-up, the mean global NMS burden (NMSS total score) significantly increased in PD patients by 18.8% (from 45.08 ± 37.62 to 53.55 ± 42.28; *p* < 0.0001; N = 501; 60.2% males, mean age 62.59 ± 8.91; mean disease duration 5.5 ± 4.37 years) compared to no change observed in controls (from 14.74 ± 18.72 to 14.65 ± 21.82; *p* = 0.944; N = 122; 49.5% males, mean age 60.99 ± 8.32) (difference between groups, *p* = 0.002) (Table 1 and Figure 1B). Moreover, the increase in the mean NMSS total score from V0 to V1 and from V1 to V2 was significant as well (*p* < 0.0001; Figure 1A). By domains, the score of all the NMSS domains at V2 was significantly higher than at V0 in PD patients except domain 9 (Miscellaneous) but none in the control group (Table 1). In PD patients, the highest increase from V0 to V2 was for domain 1 (cardiovascular symptoms) with a 117.7% (Cohen’s effect size = 0.644), and the lowest was for domain 9 (miscellaneous) with a 10.7% (Figure 1C). With regard to other NMS, a significant change in the score of the PD-CRS (from 92 ± 15.65 to 90.26 ± 18.07; *p* < 0.0001; N = 500), VAS-Pain (from 2.61 ± 2.92 to 2.96 ± 2.88; *p* = 0.037; N = 501), and VAFS-Physical (from 2.86 ± 2.67 to 3.17 ± 2.8; *p* = 0.025; N = 501) was also observed in the PD group but not in controls (Table 1).

In 201/501 patients (40.1%) a score at V2 equal to or less than the baseline NMSS total score was observed compared to in 73/122 controls (59.8%) (*p* < 0.0001). Specifically, 38.5%, 1.6%, and 59.9% of the patients presented improvement (NMSS at V2 < NMSS at V0), no change (NMSS at V2 = NMSS at V0), and impairment (NMSS at V2 > NMSS at V0), respectively, with regard to the change in the NMSB from V0 to V2 (Figure 2B). A significant change in the percentage of patients with mild, moderate, severe, or very severe burden from V0 to V1 (*p* = 0.016; N = 600) and from V1 to V2 (*p* = 0.007; N = 482) was observed in the PD group (when all groups were considered, *p* < 0.0001 [N = 482]) (Figure 2A).

A moderate correlation was observed between the change in the NMSS total score from V0 to V2 and the change in the BDI-II (r = 0.35; *p* < 0.0001), PDSS (r = −0.34; *p* < 0.0001), NPI (r = 0.31; *p* < 0.0001), and PDQ-39SI (r = 0.42; *p* < 0.0001) scores over the same period (Table 2). When the change in the score of the domains of the NMSS was considered, moderate correlations were observed between the changes from V0 to V2 in domain 1 (cardiovascular symptoms) and ADLS (r = −0.35; *p* < 0.0001), domain 2 and PDSS (r = −0.34; *p* < 0.0001) and PDQ-39SI (r = 0.33; *p* < 0.0001), and domain 3 and BDI-II (r = 0.33; *p* < 0.0001), VAS-PAIN (r = 0.41; *p* < 0.0001), and PDQ-39SI (r = 0.35; *p* < 0.0001) (Table 2). Correlation between the change from V0 to V2 in NMS (NMSS total score and score domains) and LEDD, daily dose of levodopa, and daily dose of dopamine agonist was negligible.

In the multivariate analysis, a greater increase in the NMSS total score from V0 to V2 was associated with longer disease duration (*p* = 0.006), a higher NPI total score (indicative of more severe neuropsychiatric symptoms) (*p* < 0.0001) and a lower total score on the NMSS (*p* < 0.0001) and PDSS (indicative of a worse sleep quality) (*p* = 0.001) at baseline, a greater decrease in the PDSS total score from V0 to V2 (indicative of a greater sleep quality impairment) (*p* < 0.0001), a greater increase in the FOGQ from V0 to V2 (indicative of a more worsening in gait problems) (*p* = 0.001) and in the NPI total score from V0 to V2 (indicative of a more impairment in neuropsychiatric symptoms) (*p* < 0.0001), and to be taking an antipsychotic agent at V2 (*p* = 0.003) (Table 3). NMSS total score at baseline (β = −0.52), change from V0 to V2 in PDSS (β = −0.34), and change from V0 to V2 in NPI (β = 0.25) provided the highest contributions to the model (adjusted R-squared 0.41; Durbin-Watson test = 1.865) (Table 3). In the final model, tolerance was from 0.61 to 0.94 and VIF from 1.05 to 1.72.

When change from V0 to V2 in the NMSS total score was considered as a binary variable (to be an NMS—improver as dependent variable), a lower dopamine agonist equivalent dose (OR = 0.998; 95%CI, 0.996–0.999; *p* = 0.013) and NPI total score (OR = 0.926; 95%CI, 0.884–0.970; *p* = 0.001) at V0, a higher NMSS total score (OR = 1.028; 95%CI, 1.018–1.039; *p* < 0.0001) at V0, and a greater decrease in the NPI total score (OR = 0.902; 95%CI, 0.864–0.943; *p* < 0.0001) and VASF-Physical (OR = 0.901; 95%CI, 0.831–0.977; *p* < 0.0001) from V0 to V2 were independent predictors of NMS improvement at 2-year follow-up (adjusted R-squared 0.27; Hosmer–Lemeshow test = 0.160). In NMS—improvers, the mean NPI and VASF-Physical total scores decreased from V0 to V2 in 2.34 ± 8.55 and of 0.39 ± 3.03 points, respectively, compared to an increase in 2.17 ± 8.57 and 0.74 ± 2.88 in non NMS improvers, respectively (*p* < 0.0001 for both analysis). The motor phenotype was not associated with NMS outcome, being NMS—improvers 104 out of 262 (39.7%) PD patients with a tremor-dominant motor phenotype vs. 89 out of 230 (37.2%) patients with a PIGD/indeterminate motor phenotype (*p* = 0.573).

## 4. Discussion

The present study observes that global NMS burden (defined as NMSS total score) demonstrates short-term progression in PD patients and identifies some factors associated with this progression. Specifically, worsening gait problems (FOGQ), sleep symptoms (PDSS), and neuropsychiatric symptoms (NPI) was associated with an increase in global NMS burden at 2-year follow-up after adjustment to baseline status. On the contrary, a progression in NMS burden in the control group was not observed. Awareness of the progression of these symptoms in clinical practice, with the aim to introduce interventions to reduce NMS burden progression when possible, could be important. As example, bilateral subthalamic stimulation (STN-DBS) improved significantly the global SNMS burden and QoL in advanced PD patients after 36 months but not in patients who received the standard-of-care medical therapy [19]. Since global NMS burden impacts the patient’s QoL [8] and its progression worsens QoL [13], understanding the role global NMS plays in PD is very relevant.

Although NMS are common and their importance has been increasing during the last years in PD [20], very little is known about the progression of NMS in PD patients, partially because there is a lack of longitudinal studies [12]. To our knowledge, our study represents the largest observational study about the longitudinal evolution of NMS in PD patients compared with non-PD controls in which a very extensive assessment and analysis have been conducted. Aligning with previous studies, we observed a significant increase in NMS burden over 2 years in PD patients compared with controls [9,10,11,12]. Erro et al. reported 2-year and 4-year longitudinal data about the baseline prevalence and longitudinal evolution of NMS in a cohort of PD patients, but the sample was small (N = 91 for 2-year and N = 61 for 4-year follow-up), a control group was not included, and a qualitative scale (NMSQ (Non-Motor Symptoms Questionnaire)) was used instead of a quantitative one [9,10]. As we observed, they found no association between NMS progression and motor disability as measured by the UPDRS-III and either drug class (i.e., L-dopa, DA, etc.) or total LEDD. In our study, the only motor feature related with NMS burden progression was gait problems. Previous studies have reported that gait problems including freezing of gait (FOG) are associated with higher NMSS total score and some NMS in particular such as cognitive impairment or anxiety [21,22,23]. Further, discrete gait impairment progresses in PD over time [24], as we noted here. In practice, it is important to keep in mind that patients with PD who develop an impairment in gait could progress more in their global NMS burden. Using gait impairment as a simple clinical biomarker of NMS progression could be helpful.

More recently, Simuni et al. [12], identified clinical and biological variables associated with NMS progression in 380 PD patients from the PPMI (Parkinson’s Progression Markers Initiative) and compared them with a control group (174 health control subjects) [12]. Although they used the MDS-UPDRS (Movement Disorder Society-Unified Parkinson’s Disease Rating Scale) Part I [25], a strong convergent validity was shown with the NMSS [26]. They observed a significant worsening in global cognition, depression, autonomic dysfunction, and impulse control disorders in PD subjects, while controls worsened only in cognition and trait anxiety. In our cohort, the mean score of all domains of the NMSS except domain 9 (miscellaneous) increased significantly in patients but not in controls, although a significant change from baseline to 2-year assessment was not observed in the rest of scales except for the PD-CRS and VAS for pain and physical fatigue. In general, these results align with a majority of previous reports [27,28,29] but contrary to others [30]. Antonini et al. [27] observed in 707 PD patients that sleep, gastrointestinal, attention/memory, and skin disturbances became more prevalent during a 24-month follow-up while psychiatric, cardiovascular, and respiratory disorders became less prevalent. In another study, Vu et al. [28] reported that anxiety and depression were significantly less frequent in 795 de novo PD patients, while pain, sexual difficulties, and weight change were more frequent at 2-year follow-up period. In another study, NMS severity in 117 PD patients increased after a mean follow-up visit of 21.6 ± 5.6 months [29]. Unlike the above, Prakash et al. [30] noted in 147 PD patients a small but significant median reduction of total NMS burden over a 12–18-month study period, postulating that the improvement in total NMS burden with various therapies may be attributed to the early stages of PD in a majority (more than 80%) of their study patients. The early stages of PD are known to have the best therapeutic window response to symptomatic therapy for motor symptoms and it could be the best window for NMS as well [30,31]. However, in the COPPADIS cohort more than 90% of the patients included at baseline were in stage 1 or 2 of H&Y and a progression of some symptoms impacting on NMS burden progression was still observed even after considering different treatments in the model (dopaminergic and non-dopaminergic therapies). In spite of a lack of longitudinal studies about cognitive changes in the short-term in PD patients, there seems to be a slow and heterogenic progression of cognitive function in PD [32]. A very recent study observed that PD patients had significantly more fatigue than the control group from baseline and throughout 9 years of follow-up [33]. Although there is no clear evidence about the progression of pain in PD and again, more longitudinal studies are needed, some reports suggest a higher frequency of pain after long-term progression [34]. Another study demonstrated that autonomic dysfunction is not only common in early-stage PD, but it increases in severity with increasing disease stage [35]. In addition, orthostatic hypotension has been associated with impaired daily living activities even in the asymptomatic stage [36]. In the COPPADIS cohort, cardiovascular symptoms was the domain of the NMSS with the greatest impairment, and a moderate correlation between the changes in cardiovascular symptoms and the changes in the ADLS at 2-year follow-up was observed. All these findings suggest that, despite symptomatic treatment, there is a lot of variability between patients with different non-motor phenotypes proposed [37] and that, globally, there is a slow progression of motor symptoms and NMS in PD. Results about the effect of size in this cohort suggest that the progression of NMS could be slower than motor symptoms.

Simuni et al. [12] identified only two predictors of the longitudinal change of NMS, age at the time of analysis and lower CSF Aβ1-42. In our analysis, as in some other reports [29,38,39] but unlike some cross-sectional and case–control studies [40,41], age was not related with the severity evolution of NMS in PD. The large heterogeneity of PD patients in terms of treatments and age (in our study patients older than 75 years old were not included) may account for this discrepancy; therefore, longer periods of follow-up studies are needed. On the contrary, a longer disease duration was related to a greater global NMS burden impairment, aligning with the observation that NMS become more prevalent and severe in advanced PD patients throughout the course of the disease [42]. We did not introduce MRI and CSF biomarkers. Mollenhauer et al., did it and did not identify an association of NMS with the CSF measures [43]. Unlike Simuni et al., information about the variance and β regression coefficients were provided in our analysis. A novel aspect to be highlighted of the present study is the fact that greater sleep symptoms (PDSS) and neuropsychiatric symptoms (NPI) impairment predicted a greater global NMS burden increase. The NMSS includes questions about these symptoms, but it has not been previously reported and this relationship was not observed for example with mood, a factor impacting very importantly in the perception of the patients about NMS [44]. As neuropsychiatric symptoms impact on QoL and several of these symptoms have a presumed effective treatment [45,46], doctors should be encouraged to assess the presence of these symptoms for the purpose of applying the proper treatment to improve the patients’ health-related QoL [46,47]. Remarkably, a reduction in the NPI score in this study from baseline visit to 2-year follow-up visit was a predictor of being an NMS—improver (i.e., to have a lower global NMS burden after 2 years of follow-up). With regard to sleep problems, there is a progressive increase in the frequency of sleep disturbances in PD, with the number of subjects reporting multiple sleep disturbances increasing over time [48]. Recently, we observed in the baseline analysis of the COPPADIS cohort that NMS burden was associated with sleep problems after adjustment for age, gender, disease duration, LEDD, H&Y, UPDRS-III, UPDRS-IV, PD-CRS, BDI-II, NPI, VAS-Pain, VAFS, FOGQ, and total number of non-antiparkinsonian treatments [49]. Here, after a 2-year follow-up we observed that changes in sleep predict changes in the global NMS burden. A large number of patients are not treated for their sleep disturbances [48] and these findings call for an increased awareness of sleep problems in PD patients. Finally, the most significant predictor of global NMS burden progression was the NMSS total score at baseline. The risk is higher when the score is lower, signifying that even in early PD patients it is necessary to be alert when the patient has a low global NMS burden because it is liable to increase. Again, it is logical to consider and on the contrary, a greater improvement in NMS after an intervention has been observed in patients with a greater NMS burden [50], so changes are related to the score at baseline.

The present study has some limitations. First, the information about NMS burden follow-up was recorded in 501 patients of 677 (74%). Thirty-eight patients dropped out of the study (1 death; 2 with change in diagnosis; 35 other reasons) at the 2-year follow-up and 132 were not evaluated. In six patients, NMSS was not assessed. However, this is a limitation observed in other prospective studies. In the study of Antonini et al. [27], 707 PD patients from 1142 initially included (61.9%) were evaluable at 24 months. The percentage in other studies was 89.8% (380/423) [12], 83.6% (117/140) [29], and 73.9% (147/199) [30]. Second, our sample was not fully representative of the PD population due to inclusion and exclusion criteria (i.e., age limit, no dementia, no severe comorbidities, no second line therapies, etc.) [15], which leads to an early PD bias in this cohort. Third, for some variables, the information was not collected in all cases. Fourth, a specific tool for assessing comorbidity, like Charlson Index or others, has not been used. However, the total number of non-antiparkinsonian medications has been suggested as a useful marker of comorbidity in PD [8]. Fifth, the smaller sample size of the control group. On the contrary, the strengths of our study include the large sample size of the PD group, a very thorough assessment, a prospective longitudinal follow-up design, the fact that this analysis was “a priori” planned as one objective of the multicenter COPPADIS project [15], and the extensive clinical and demographic information recorded.

In conclusion, the present study observes that global NMS burden demonstrates short-term progression in PD patients but not in controls and identifies worsening sleep problems and neuropsychiatric symptoms as significant independent predictors of this NMS progression. Strategies designed to act over these symptoms and to analyze the short-term and long-term progression of NMS burden could be of interest.

## Figures and Tables

**Figure 1 jpm-11-00626-f001:**
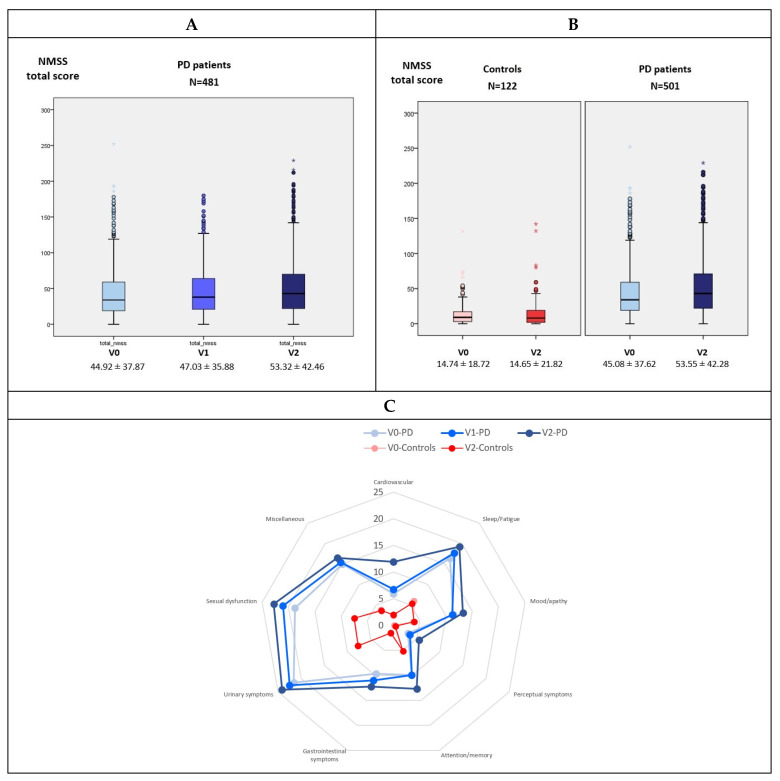
(**A**) Mean NMSS total score at V0 (baseline), V1 (1 year follow-up ± 1 month), and V2 (2-year follow-up ± 1 month) in PD patients (N = 481). V1 vs. V0, *p* < 0.0001; V2 vs. V1, *p* < 0.0001; V2 vs. V0, *p* < 0.0001; All groups; *p* < 0.0001. (**B**) Change from V0 to V2 in the mean NMSS total score in PD patients (N = 501) and controls (N = 122). V2 vs. V0 in controls, *p* = 0.944; V2 vs. V0 in PD patients, *p* < 0.0001; difference in the change from V0 to V2 between PD patients and controls, *p* = 0.002. (**C**) Mean score on each domain of the NMSS scale at V0, V1, and V2 in PD patients (blue) and controls (red). Data (A and B) are presented as box plots, with the box representing the median and the two middle quartiles (25–75%). *P* values were computed using general linear models (GLM) repeated measures. Mild outliers (O) are data points that are more extreme than Q1 − 1.5 × IQR or Q3 + 1.5 × IQR. NMSS, Non-Motor Symptoms Scale.

**Figure 2 jpm-11-00626-f002:**
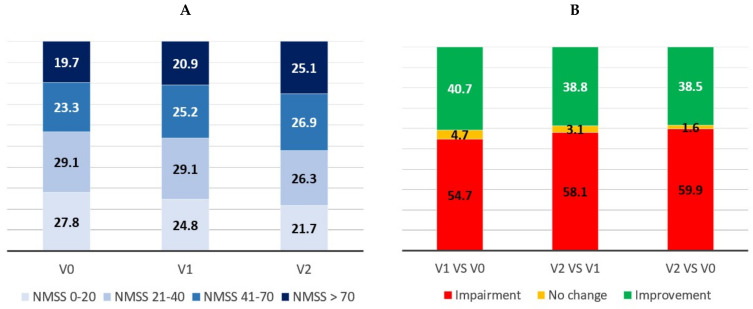
(**A**) Percentage of PD patients with mild (NMSS 1–20), moderate (NMSS 21–40), severe (NMSS 41–70), and very severe (NMSS >70) NMS burden at V0 (baseline), V1 (1 year follow-up ± 1 month), and V2 (2-year follow-up ± 1 month). V1 vs. V0, *p* < 0.016 (N = 600); V2 vs. V1, *p* = 0.007 (N = 482), All groups; *p* < 0.0001 (N = 481). (**B**) Percentage of patients with a greater NMS burden (NMS burden impairment; in red), no change (No changes in the NMSS total score), and a lower NMS burden (NMS burden improvement; in green) at V1 compared to V0, V2 compared to V1, and V2 compared to V0. NMSS, Non-Motor Symptoms Scale.

**Table 1 jpm-11-00626-t001:** Changes in motor and non-motor symptoms, disability, and quality of life in PD patients and/or controls from V0 (baseline) to V2 (2 years ± 1 month).

	PD Patients V0	PD Patients V2	Cohen’s Test	*p* ^a^	Controls V0	Controls V2	*p* ^b^	*p* ^c^	*p* ^d^
Hoehn & Yahr (OFF) (%)			0.422	<0.0001	N.A.	N.A.	N.A.	N.A.	N.A.
Stage 1	22.7	13.3							
Stage 2	68	77							
Stage 3–5	9.3	9.7							
UPDRS-III (OFF)	21.92 ± 10.53	25.26 ± 12.19	0.452	<0.0001	N.A.	N.A.	N.A.	N.A.	N.A.
UPDRS-IV	1.99 ± 2.41	2.65 ± 2.75	0.371	<0.0001	N.A.	N.A.	N.A.	N.A.	N.A.
FOGQ	3.76 ± 4.69	4.94 ± 5.18	0.4	<0.0001	N.A.	N.A.	N.A.	N.A.	N.A.
Daily dose L-dopa (mg)	577.48 ± 412.09	767.56 ± 307.1	0.812	<0.0001	N.A.	N.A.	N.A.	N.A.	N.A.
Number of non-antipark. drugs	2.35 ± 2.38	3.08 ± 2.65	0.507	<0.0001	2.04 ± 2.16	2.76 ± 2.35	0.005		
PD-CRS	92 ± 15.65	90.26 ± 18.07	−0.207	<0.0001	99.65 ± 13.56	99.68 ± 13.73	0.368	0.517	0.018
NMSS	45.08 ± 37.62	53.55 ± 42.28	0.343	<0.0001	14.74 ± 18.72	14.65 ± 21.82	0.944	0.784	0.002
Cardiovascular	5.45 ± 9.79	11.87 ± 14.21	0.644	<0.0001	1.78 ± 3.6	1.99 ± 3.91	0.929	0.151	0.004
Sleep/fatigue	16.28 ± 15.91	19.2 ± 17.52	0.241	<0.0001	5.91 ± 7.98	5.32 ± 9.87	0.813	0.243	0.018
Mood/apathy	11.32 ± 16.07	13.21 ± 17.98	0.173	0.007	3.8 ± 8.85	3.93 ± 11.53	0.219	0.438	0.249
Perceptual symptoms	3.24 ± 9.24	5.51 ± 12.71	0.286	<0.0001	0.13 ± 1.12	0.39 ± 2.41	0.352	0.643	0.316
Attention/memory	10.01 ± 14.5	12.69 ± 17.08	0.249	<0.0001	5.13 ± 10.85	5.18 ± 11.25	0.567	0.747	0.321
Gastrointestinal symptoms	9.59 ± 13.1	12.28 ± 14.35	0.3	<0.0001	1.58 ± 5.23	1.53 ± 4.39	0.385	0.414	0.001
Urinary symptos	21.47 ± 22.21	24.09 ± 23.14	0.181	0.013	7.63 ±12.57	7.71 ±11.74	0.851	0.982	0.012
Sexual dysfunction	18.83 ± 25.53	22.67 ± 27.84	1.182	0.005	7.43 ± 16.54	7.5 ± 14.68	0.903	0.714	0.002
Miscellaneous	14.98 ± 15.41	16.59 ± 15.69	0.149	0.054	3.73 ± 8.75	3.54 ± 8.31	0.633	0.698	0.004
BDI-II	8.28 ± 6.9	8.54 ± 7.48	0.048	0.369	4.56 ± 5.46	4.31 ± 5.5	0.484	0.541	0.043
PDSS	117.13 ± 24.48	117.85 ± 24.98	0.038	0.452	131.26 ± 17.41	126.67 ± 26.46	0.796	0.737	0.677
QUIP-RS	4.6 ± 8.8	4.66 ± 9.22	0.007	0.798	1.51 ± 3.73	1.32 ± 3.37	0.284	0.334	0.222
NPI	5.82 ± 7.88	6.17 ± 9.39	0.056	0.491	3.31 ± 7.15	2.64 ± 7.67	0.786	0.714	0.486
VAS-PAIN	2.61 ± 2.92	2.96 ± 2.88	0.147	0.037	1.49 ± 2.41	1.70 ± 2.32	0.272	0.661	0.177
VASF-physical	2.86 ± 2.67	3.17 ± 2.8	0.147	0.025	1.52 ± 2.35	1.29 ± 2.12	0.78	0.488	0.198
VASF-mental	2.09 ± 2.51	2.20 ± 2.61	0.056	0.545	1.29 ± 2.09	1.03 ± 1.97	0.699	0.96	0.254
ADLSL	88.58 ± 10.19	84.26 ± 13.38	−0.519	<0.0001	98.87 ± 6.65	99.52 ± 2.15	0.227	0.069	<0.0001
PDQ-39SI	16.72 ± 13.02	20.3 ± 16.41	0.413	<0.0001	N.A.	N.A.	N.A.	N.A.	N.A.
PQ-10	3.77 ± 0.54	3.75 ± 0.58	−0.114	0.063	8.07 ± 1.22	7.86 ± 1.65	0.06	0.486	0.831
EUROHIS-QOL8	3.8 ± 0.7	3.68 ± 0.67	0.048	0.4	4.18 ± 0.5	4.12 ± 0.51	0.066	0.852	0.074

*p* values were computed using general linear models (GLM) repeated measures. The results represent mean ± SD or %; *p* ^a^, change over time (V2 vs. V0) in PD patients; *p* ^b^, change over time (V2 vs. V0) in controls. Age, gender and LEDD (levodopa equivalent daily dose) (except for assessing changes in this variable) at V0 and at V2 were included as covariates; *p* ^c^, group visit interaction; *p* ^d^, PD vs. controls. PD vs. controls is not applicable if test of interaction was significant (a significant test of interaction means the rates of changes over time are different between the two groups). All patients with the data at V0 and V2 were included for each comparative analysis. For NMSS and its domains, N = 501 in PD patients and N = 122 in the control group. ADLS, Schwab & England Activities of Daily Living Scale; BDI-II, Beck Depression Inventory-II; FOGQ, Freezing Of Gait Questionnaire; NMSS, Non-Motor Symptoms Scale; NPI, Neuropsychiatric Inventory; PD-CRS, Parkinson’s Disease Cognitive Rating Scale; PDSS, Parkinson’s Disease Sleep Scale; QUIP-RS, Questionnaire for Impulsive-Compulsive Disorders in Parkinson’s Disease-Rating Scale; UPDRS, Unified Parkinson’s Disease Rating Scale; VAFS, Visual Analog Fatigue Scale; VAS-Pain, Visual Analog Scale-Pain.

**Table 2 jpm-11-00626-t002:** Correlations between the changes in non-motor symptoms (NMSS total score and NMSS domains) and other disease-related variables in PD patients from V0 (baseline) to V2 (2 years ± 1 month).

	NMSS TS	NMSS D1	NMSS D2	NMSS D3	NMSS D4	NMSS D5	NMSS D6	NMSS D7	NMSS D8	NMSS D9
Age at baseline	−0.06	0.12 ***	−0.03	−0.01	0.04	0.04	−0.04	−0.08	−0.03	−0.04
Disease duration (at V0)	0.10 ***	0.11 ***	0.02	0.07	0.06	0.04	0.02	0.06	0.12 ***	0.05
N. of non-antipark. drugs (at V0)	−0.05	0.12 ***	−0.03	−0.03	0	−0.02	−0.03	−0.06	−0.02	−0.07
Change at V2 (from V0 to V2)										
LEDD	0.02	−0.02	0.03	0.02	0.02	0.01	0.08	0.04	−0.01	0
Daily dose L-dopa (mg)	0.01	−0.05	0.01	0.03	−0.02	0.02	0.05	0.01	−0.03	0.04
Equivalent daily dose of DA (mg)	−0.04	0.02	0.02	−0.1	0.02	−0.09 ***	0.05	−0.04	−0.06	−0.05
Number of non-antipark. drugs	0.06	0.07	−0.02	0	0.09 ***	0.08	0.03	0.07	0	0.04
UPDRS-III (OFF)	0.16 *	0.07	0.12 ***	0.16 **	0.07	0.11 ***	0.07	0.14 ***	0	0.15 ***
UPDRS-IV	0.10 ***	0.02	0.12 ***	0.09	0.06	0	0.06	0.11 ***	0	0.04
FOGQ	0.21 *	0.11 ***	0.19 *	0.13 **	0.09 ***	0.13 ***	0.12 ***	0.14 ***	0.10 ***	0.11 ***
PD-CRS	0.01	−0.05	−0.04	−0.01	−0.05	−0.11 ***	0.02	−0.03	0.15	0.03
BDI-II	0.35 *	0.17 *	0.29 *	0.33 *	0.19 *	0.22 *	0.09 ***	0.15 **	0.13 ***	0.14 ***
PDSS	−0.34 *	−0.04	−0.34 *	−0.23	−014 **	−0.15 **	−0.17*	−0.24 *	−0.12 ***	−0.17 *
QUIP-RS	0.10 ***	0	0.06	−0.03	0.17 *	0.05	−0.02	0.11 ***	0.03	0.05
NPI	0.31 *	0.12 ***	0.19 *	−0.03	0.21 *	0.18 *	0.12 ***	0.12 ***	0.14 ***	0.06
VAS-PAIN	0.12 ***	0.09 ***	0.09 ***	0.41 *	0.06	0.01	0.07	0.02	0.11 ***	0.12 ***
VASF-physical	0.21 *	0.01	0.10 ***	0.11 ***	0.07	0.08	0.13 ***	0.08	0.09	0.14 ***
VASF-mental	0.21 *	0	0.15 *	0.18 *	0.07	0.11	0.18 *	0.09	0.07	0.10 ***
PDQ-39SI	0.42 *	0.09 ***	0.33 *	0.35 *	0.15 **	0.23 *	0.20 *	0.19 ***	0.12 ***	0.17 *
PQ-10	−0.17 *	−0.06	−0.15 **	−0.17 *	−0.11 ***	−0.11 ***	−0.07	−0.07	−0.01	−0.08
EUROHIS-QOL8	−0.20 *	−0.03	−0.25 *	−0.09 ***	−0.11 ***	−0.12 ***	−0.05	−013 ***	−0.04	−0.09
ADLS	−0.24 *	−0.35 *	−0.19 *	−0.19	−0.15 **	−0.18 *	−0.17	−0.15 **	−0.17 *	−0.10 ***

Spearman correlation test was applied. *, *p* < 0.0001; **, *p* < 0.001; ***, *p* < 0.05. In bold are expressed significant values. NMSS: TS, total score. D1, Cardiovascular (items 1 and 2; score, 0 to 24); D2, Sleep/fatigue (items 3, 4, 5, and 6; score, 0 to 48); D3, Depression/apathy (items 7, 8, 9, 10, 11, and 12; score, 0 to 72); D4, Perceptual problems/hallucinations (items 13, 14, and 15; score, 0 to 36); D5, Attention/memory (items 16, 17, and 18; score, 0 to 36); D6, Gastrointestinal tract (items 19, 20, and 21; score 0 to 36); D7, Urinary symptoms (items 22, 23, and 24; score, 0 to 36); D8, Sexual dysfunction (items 25 and 26; score 0 to 24); D9, Miscellaneous (items 27, 28, 29, and 30; score, 0 to 48). ADLS, Schwab & England Activities of Daily Living Scale; BDI-II, Beck Depression Inventory-II; DA, dopamine agonist; FOGQ, Freezing Of Gait Questionnaire; LEDD, levodopa equivalent dauly dose; N, number; NMSS, Non-Motor Symptoms Scale; NPI, Neuropsychiatric Inventory; PD-CRS, Parkinson’s Disease Cognitive Rating Scale; PDSS, Parkinson’s Disease Sleep Scale; QUIP-RS, Questionnaire for Impulsive-Compulsive Disorders in Parkinson’s Disease-Rating Scale; TS, total score; UPDRS, Unified Parkinson’s Disease Rating Scale; VAFS, Visual Analog Fatigue Scale; VAS-Pain, Visual Analog Scale-Pain.

**Table 3 jpm-11-00626-t003:** Linear regression model about factors associated with global burden progression after 2-year follow-up (change in the NMSS total score from V0 to V2).

	β ^a^	β ^b^	95% IC ^a^	95% IC ^b^	*p* ^a^	*p* ^b^
**At V0 (baseline)**						
Age at baseline	−0.036	N.A.	−0.501–0.211	N.A.	0.423	N.A.
Gender	−0.045	N.A.	−9.419–3.024	N.A.	0.313	N.A.
Disease duration	0.092	0.118	0.019–1.499	0.261–1.591	0.044	0.006
Number of non-antipark. drugs	−0.041	N.A.	−1.862–0.683	N.A.	0.363	N.A.
LEDD	0.062	N.A.	−0.002–0.012	N.A.	0.17	N.A.
Daily dose L-dopa (mg)	0.077	0.152	−0.001–0.019	N.A.	0.085	N.A.
Equivalent Daily dose of DA (mg)	0.08	N.A.	−0.002–0.038	N.A.	0.075	N.A.
No-tremoric motor phenotype	0.062	0.094	−1.788–10.373	0.675–12.195	0.166	0.029
UPDRS-III	0.026	N.A.	−0.217–0.387	N.A.	0.58	N.A.
UPDRS-IV	0.015	N.A.	−1.047–1.473	N.A.	0.74	N.A.
FOGQ	−0.023	N.A.	−0.823–0.482	N.A.	0.608	N.A.
PD-CRS	−0.03	N.A.	−0.262–0.129	N.A.	0.504	N.A.
NMSS	−0.317	−0.52	−0.369–0.215	−0.557–−0.370	<0.0001	<0.0001
BDI-II	−0.174	N.A.	−0.872	N.A.	<0.0001	N.A.
PDSS	0.077	−0.186	−0.015–0.234	−0.397–−0.106	0.085	0.001
NPI	−0.12	0.208	−0.912–0.107	0.441–1.353	0.013	<0.0001
QUIP-RS	−0.005	N.A.	−0.310–0.271	N.A.	0.896	N.A.
VAS-PAIN	0.032	N.A.	−0.62–0.148	N.A.	0.032	N.A.
VASF-Physical	−0.021	N.A.	−1.181– -0.684	N.A.	0.601	N.A.
VASF-Mental	−0.015	N.A.	−1.206–0.832	N.A.	0.719	N.A.
**Change at V2 (from V0 to V2)**						
Number of non-antipark. drugs	0.087	N.A.	−0.040–3.925	N.A.	0.055	N.A.
LEDD	0.072	N.A.	−0.002–0.017	N.A.	0.115	N.A.
Daily dose L-dopa (mg)	0.044	N.A.	−0.007–0.019	N.A.	0.339	N.A.
Equivalent daily dose of DA (mg)	0.009	N.A.	−0.013–0.016	N.A.	0.843	N.A.
UPDRS-III (OFF)	0.178	N.A.	0.296–0.938	N.A.	<0.0001	N.A.
UPDRS-IV	0.086	N.A.	−0.041–2.434	N.A.	0.058	N.A.
FOGQ	0.284	0.149	1.658–3.074	0.522–1.922	<0.0001	0.001
PD-CRS	−0.026	N.A	−0.339–0.186	N.A.	0.567	N.A.
BDI-II	0.334	N.A.	1.120–1.862	N.A.	<0.0001	N.A.
PDSS	−0.303	−0.339	−0.222	−0.543–−0.292	<0.0001	<0.0001
NPI	0.307	0.249	0.798–1.521	0.595–1.296	<0.0001	<0.0001
QUIP-RS	0.095	N.A.	0.038–0.612	N.A.	0.027	N.A.
VAS-PAIN	0.156	N.A.	0.761–2.302	N.A.	<0.0001	N.A.
VASF-Physical	0.192	0.101	1.258–2.973	0.179–2.061	<0.0001	0.02
VASF-Mental	0.194	N.A.	1.282–3.010	N.A.	<0.0001	N.A.
**At V2**						
To be taking L-dopa	0.025	N.A.	−7.129–12.723	N.A.	0.58	N.A.
To be taking a MAO-B inhibitor	0.03	N.A.	−4.722–9.684	N.A.	0.499	N.A.
To be taking a COMT inhibitor	0.003	N.A.	−6.564–6.987	N.A.	0.951	N.A.
To be receiving a DA	0.067	N.A.	−1.651–12.057	N.A.	0.136	N.A.
To be taking an analgecic	−0.074	N.A.	−11.116–0.388	N.A.	0.068	N.A.
To be taking a benzodiazepin	0.025	N.A.	−4.350–8.277	N.A.	0.542	N.A.
To be taking an antidepressive agent	0.051	N.A.	−2.049–9.514	N.A.	0.205	N.A.
To be taking an antipsychotic agent	0.067	0.129	6.067–29.381	−3.084–22.227	0.138	0.003
To practice regular exercise	0.011	N.A.	−6.668–8.440	N.A.	0.818	N.A.
Cognitive stimulation therapy	−0.017	N.A.	−7.935–5.396	N.A.	0.708	N.A.
Physiotherapy	0.016	N.A.	−5.661–8.179	N.A.	0.721	N.A.
Speech therapy	−0.008	N.A.	−10.296–8.548	N.A.	0.885	N.A.

Dependent variable: change from V0 to V2 in the NMSS total score. β standardized coefficient and 95% IC are shown. ^a^, univariate analysis; ^b^, multivariate analysis (Durbin-Watson test = 1.865; R^2^ = 0.41). BDI-II, Beck Depression Inventory-II; COMT, catechol-o-methyl transferase; DA, dopamine agonist; FOG, freezing of gait; FOGQ, Freezing Of Gait Questionnaire; LEDD, levodopa equivalent daily dose (mg); MAO-B, monoamine oxidase type B; N.A., not applicable; NMS, non-motor symptoms; NMSS, Non-Motor Symptoms Scale; NPI, Neuropsychiatric Inventory; PD, Parkinson’s disease; PD-CRS, Parkinson’s Disease Cognitive Rating Scale; PDQ-39SI, 39-item Parkinson’s Disease Quality of Life Questionnaire Summary Index; PDSS, Parkinson’s Disease Sleep Scale; QoL, Quality of life; QUIP-RS, Questionnaire for Impulsive-Compulsive Disorders in Parkinson’s Disease-Rating Scale; UPDRS, Unified Parkinson’s Disease Rating Scale; VAFS, Visual Analog Fatigue Scale; VAS-Pain, Visual Analog Scale-Pain.

## Data Availability

The protocol and the statistical analysis plan are available on request. Deidentified participant data are not available for legal and ethical reasons.

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
