# Peer review of "Predictors of Global Non-Motor Symptoms Burden Progression in Parkinson’s Disease. Results from the COPPADIS Cohort at 2-Year Follow-Up"

_jpm, 2021, doi:10.3390/jpm11070626_

Round 1
Reviewer 1 Report
This manuscript from an experienced group presents the results of a large cohort of PD patients regarding NMS burden progression. Although the results are already relevant for the clinician, I think that this data has the potential for some more novelty (see suggestions below). Additionally, I have some concerns regarding the statistical analysis which should be addressed.
Major comments:
Methods:
- There is no correction for multiple testing included in the statistics. Regarding the mass of compared scales this seems necessary. This might also resolve the “significance” in change in perceptual symptoms in the control group and some other scales in the PD group.
- When comparing V0 to V1 to V2 in PD patients the authors only conduct post-hoc tests without an appropriate statistic for comparison of multiple groups.
- Following Figure 1 a total of 176 patients dropped out of the NMSS assessment at V2. If I get it right (Fig 1A), the number of patients differed between the assessments. As the aim of the study was to determine the NMS progression, I would suggest a supplemental analysis only including patients with complete datasets, as the drop outs might lead to a bias of the progression effect.
Discussion:
- It remains unclear to the reader how the authors chose the respective factors to be tested for inclusion in the regression model. In my opinion it would also be interesting to include H&Y at baseline, change in H&Y, BDI-II at baseline and change in BDI-II and PDQ-39 at baseline and change in PDQ-39. Especially as the authors state in the disuccions that no relationship was observed with mood but did not include the BDI-II in their model.
- Besides the finding that total NMS burden significantly increases over 2 years another interesting finding is that ~40% of the patients improve! It would be really interesting to discuss this more: i) Where there some NMSS domains which were more likely to improve and others that were more likely to worsen? ii) which factors led to an improvement of these domains (increase in LEDD, DA, other drugs added? Social factors? etc). When analyzing DBS outcomes we sometimes refer to responder and non-responder regarding specific NMS outcomes (see e.g. 1016/j.brs.2018.03.009). Just as an idea: One might also think of dichotomizing the cohort in “NMS-improver” and “NMS-worsener” and investigate those more specifically. This also leads to my next point:
- As also discussed by the authors there are already several studies reporting 2-year NMS outcomes in PD patients. I agree with the authors, that this study has one of the largest sample sizes and a control group. However, I am missing some novelty and I have one suggestion to increase the novelty of this paper.
In my opinion it would be really interesting to use the large sample size and dive into a subgroup analysis comparing (A): NMS progression for different PD motor subtypes and, maybe more challenging (B), comparing NMS progression in different NMS subtypes as e.g. described by Sauerbier et al. 2015 or Mu et al. 2017. The question is: Is the individual PD symptom profile a predictor of NMS progression?
Figure 1:
- As this is the main figure of the paper, I suggest to optimize the data visualization. i) significance of the results should be indicated by lines and stars between the respective boxplots as usual, ii) for Fig 1B I suggest not to change the outline of the boxplots but the filling as in Fig 1A, iii) just as a suggestion: to show the progression of the main outcome parameter in a cohort I really like raincloudplots as e.g. used in https://doi.org/10.1007/s13311-021-01057-y (Figure 1). Iiii) for part C I would replace the domain numbers by the domain names
Minor comments:
Introduction:
- Please don’t refer to unpublished and non-accessible research articles (Ref 13 and its referring sentence)
Methods:
- It is not indicated whether the MedOFF or the MedON UPDRSIII was considered for the regression model for patients with motor fluctuations
- I suggest introducing the concept of mild, moderate, severe, or very severe NMS burden in the Methods section. Is there any reference for the chosen differentiation?
Results:
- Please indicate the disease duration at study inclusion
Discussion:
- Do you have any thoughts on why the PDSS does not change in the PD cohort but change in PDSS is one predictor of NMS progression?
- As stated by the authors, one of the limitations of the study is its early PD bias. I think it would be interesting to add a comparison of the results in this cohort to the results of a published advanced PD cohort (e.g. MED control group at 36 month follow up in Jost et al. 2020)
Table 1:
- Please indicate the effect size (e.g. cohen’s d) for each scale investigated in the PD group
- When the V1 data is included in the analysis (which is IMO not absolutely necessary), the results should also be reported in Table 1.
Figure 2
- I suggest to indicate significant changes in the figure and not as text below it (A)
I recommend a language editing of the manuscript as there are several typos, missing words, or results reported twice in one sentence, in the manuscript and the tables (also some Abbreviations are missing below the tables).
Author Response
Reviewer #1 (Comments to the Author):
This manuscript from an experienced group presents the results of a large cohort of PD patients regarding NMS burden progression. Although the results are already relevant for the clinician, I think that this data has the potential for some more novelty (see suggestions below). Additionally, I have some concerns regarding the statistical analysis which should be addressed.
Major comments:
Methods:
There is no correction for multiple testing included in the statistics. Regarding the mass of compared scales this seems necessary. This might also resolve the “significance” in change in perceptual symptoms in the control group and some other scales in the PD group.
When comparing V0 to V1 to V2 in PD patients the authors only conduct post-hoc tests without an appropriate statistic for comparison of multiple groups.
AUTHORS – Thank you very much for your comment. As you suggest, we included GLM repeated measures to test the change in different variables over time separately in each group and test for differences between groups over time. We added it in Methods: “General linear model (GLM) repeated measure was used to test whether the mean differences of the NMSS total score and the NMSS domain scores between the two visits (V0 and V2) were significant in both PD patients and controls. In PD patients, it was analyzed the change between V1 and V0, V2 and V1 and between all groups as well. Age, gender and LEDD (levodopa equivalent daily dose) at V0 and at V2 were included as covariates. This test was also applied for testing the difference between V2 and V0 in other variables”.
Following Figure 1 a total of 176 patients dropped out of the NMSS assessment at V2. If I get it right (Fig 1A), the number of patients differed between the assessments. As the aim of the study was to determine the NMS progression, I would suggest a supplemental analysis only including patients with complete datasets, as the drop outs might lead to a bias of the progression effect.
Figure 1A shows the score on the NMSS scale at baseline (V0), at V1 and at V2 considering all patients recruited and followed. However, in the analysis of table 1, only patients with the value at V0 and at V2 (both scores) were considered for doing the comparison. This fact explains why the mean value of the total score of the NMSS is not exactly the same, although there are practically no differences. So, as you indicate, in the analysis only patients with complete datasets were included (Table 1). In fact, this is explained clearly in the text below the table as “All patients with the data at V0 and V2 were included for each comparative analysis”. In Figure 1A the value considering all patients is shown but the sample (N) is indicated according to the comparison: V1 vs V0; p<0.0001 (N=600); V2 vs V1; p<0.0001 (N=482); V2 vs V0; p<0.0001 (N=0.501); All groups; p<0.0001 (N=481).
Discussion:
It remains unclear to the reader how the authors chose the respective factors to be tested for inclusion in the regression model. In my opinion it would also be interesting to include H&Y at baseline, change in H&Y, BDI-II at baseline and change in BDI-II and PDQ-39 at baseline and change in PDQ-39. Especially as the authors state in the disuccions that no relationship was observed with mood but did not include the BDI-II in their model.
AUTHORS – Thank you very much for your comment. Obviously, this is an important point in this manuscript and we carefully described in Methods how linear regression models were conducted:
“Linear regression models were used for determining predictive factors of global NMS burden progression (NMSS total score change from V0 to V2 as dependent variable). Any variable with univariate associations with p-values < 0.20 were included in a multivariable model, and a backwards selection process was used to remove variables individually until all remaining variables were significant at the 0.10 level [12]. The variables considered for the analysis were: 1) at baseline (V0): age, gender, disease duration, total number of non-antiparkinsonian drugs, LEDD, daily dose of levodopa, daily dose of dopamine agonist, motor phenotype [18], UPDRS-III, UPDRS-IV, FOGQ, PD-CRS, NMSS, PDSS, NPI, QUIP-RS, VAS-PAIN, VASF-Physical, and VASF-Mental; 2) at 2-year follow-up (V2): to be receiving L-dopa, a MAO-B (monoamine oxidase type B) inhibitor, a COMT (catechol-o-methyl transferase) inhibitor, a dopamine agonist, and to practice regular exercise; 3) At the end of the follow-up (V2): change from V0 to V2 in total number of non- antiparkinsonian drugs, LEDD, daily dose of levodopa, daily dose of dopamine agonist, UPDRS- III, UPDRS-IV, FOGQ, PD-CRS, PDSS, NPI, QUIP-RS, VAS-PAIN, VASF-Physical, and VASF- Mental. Tolerance and variance inflation factor (VIF) were used to detect multicollinearity (multicollinearity was considered problematic when tolerance was less than 0.2 and, simultaneously, the value of VIF 10 and above). A level of p=0.01 was considered for determining significant variables in the model”.
This methodology was used for example in a very recent paper determining predictors of change in NMS at 2 years of follow-up in PD patients from the PPMI: Simuni T, Caspell-Garcia C, Coffey CS, et al. Baseline prevalence and longitudinal evolution of non-motor symptoms in early Parkinson's disease: the PPMI cohort. J Neurol Neurosurg Psychiatry 2018;89:78-88.
Therefore, the UPDRS-III was included instead of H&Y and BDI-II at V0 and the change in BDI-II score from V0 to V2 were included in the model. The PDQ-39 was not considered because it really measures the perception of quality of life as consequence of the complications of PD, that is, it is a consequence and not a cause.
After an internal review by the COPPADIS Study Group, in table 3 and with the idea of simplifying and facilitating the interpretation, only the significant variables were shown. However, as you suggest, we provide Table 3 as a complete table with all the information. Importantly, many variables were considered for the linear regression models and as it was described in Methods, multicollinearity was ruled out.
Besides the finding that total NMS burden significantly increases over 2 years another interesting finding is that ~40% of the patients improve! It would be really interesting to discuss this more: i) Where there some NMSS domains which were more likely to improve and others that were more likely to worsen? ii) which factors led to an improvement of these domains (increase in LEDD, DA, other drugs added? Social factors? etc). When analyzing DBS outcomes we sometimes refer to responder and non-responder regarding specific NMS outcomes (see e.g. 1016/j.brs.2018.03.009). Just as an idea: One might also think of dichotomizing the cohort in “NMS-improver” and “NMS-worsener” and investigate those more specifically. This also leads to my next point:
As also discussed by the authors there are already several studies reporting 2-year NMS outcomes in PD patients. I agree with the authors, that this study has one of the largest sample sizes and a control group. However, I am missing some novelty and I have one suggestion to increase the novelty of this paper.
In my opinion it would be really interesting to use the large sample size and dive into a subgroup analysis comparing (A): NMS progression for different PD motor subtypes and, maybe more challenging (B), comparing NMS progression in different NMS subtypes as e.g. described by Sauerbier et al. 2015 or Mu et al. 2017. The question is: Is the individual PD symptom profile a predictor of NMS progression?
AUTHORS – Thank you very much for your comment. In this study, the NMSS is considered for analyzing the changes as a continuous variable. In general, it is more accurate to use the NMSS and its changes over time as a continuous variable because the changes occur gradually and it is a quantitative variable. An example is the cognitive changes that occur in dementia, which are progressive and not really jumping in stages as we tend to internalize. In any case, as you suggest, we included a binary regression model to identify predictors of being a NMS - improver. The results were quite similar to the linear one (i.e., NPI, NMSS at V0, not BDI-II, etc.), highlighting the baseline NMSS total score and the change in the NPI from V0 to V2.
In the text in Methods we added: “Finally, according to the change in the NMSS total score, patients were classified as “NMS – improver” (NMSS total score at V2 < NMSS total score at V0) or “non NMS improver” (NMSS total score at V2 ≥ NMSS total score at V0) and predictors of NMS improvement were identified. A level of p=0.01 was considered for determining significant variables in the model”.
In Results, this information was added:
“When change from V0 to V2 in the NMSS total score was considered as a binary variable (to be a NMS – improver as dependent variable), a lower dopamine agonist equivalent dose (OR=0.998; 95%CI, 0.996 – 0.999; p=0.013) and NPI total score (OR=0.926; 95%CI, 0.884 – 0.970; p=0.001) at V0, a higher NMSS total score (OR=1.028; 95%CI, 1.018 – 1.039; p<0.0001) at V0, and a greater decrease in the NPI total score (OR=0.902; 95%CI, 0.864 – 0.943; p<0.0001) and VASF-Physical (OR=0.901; 95%CI, 0.831-0.977; p<0.0001) from V0 to V2 were independent predictors of NMS improvement at 2-year follow-up (adjusted R-squared 0.27; Hosmer-Lemeshow test = 0.160). In NMS – improvers, the mean NPI and VASF-Physical total scores decreased from V0 to V2 in 2.34 ± 8.55 and of 0.39 ± 3.03 points, respectively, compared to an increase in 2.17 ± 8.57 and 0.74 ± 2.88 in non NMS – improvers, respectively (p<0.0001 for both analysis)”.
Motor Phenotype was included in both models (it was previously explained in Methods) and it was not a significant predictor or NMSS total score changes. Moreover, it was not observed an association between to be a NMS – improver and motor phenotype. We provide this information in results: “The motor phenotype was not associated with NMS outcome, being NMS – improvers 104 out of 262 (39.7%) PD patients with a tremor-dominant motor phenotype vs 89 out of 230 (37.2%) patients with a PIGD/indeterminate motor phenotype (p=0.573)”. With regard to NMS phenotype, this variable has not been recorded in this study. We could consider in the future a post-hoc analysis considering NMS phenotype classification according to NMS information collected and suggestions of the literature.
Figure 1:
As this is the main figure of the paper, I suggest to optimize the data visualization. i) significance of the results should be indicated by lines and stars between the respective boxplots as usual, ii) for Fig 1B I suggest not to change the outline of the boxplots but the filling as in Fig 1A, iii) just as a suggestion: to show the progression of the main outcome parameter in a cohort I really like raincloudplots as e.g. used in https://doi.org/10.1007/s13311-021-01057-y (Figure 1). Iiii) for part C I would replace the domain numbers by the domain names
AUTHORS – Thank you very much for your comment. In figure 1A, the results of all analysis is with p<0.0001, so the information is provided in the legend. Really, we think that use lines and stars between the respective boxplots is not necessary. This information is also provided in the legend for Figure 1:
“Figure 1. A. Mean NMSS total score at V0 (baseline), V1 (1 year follow-up ± 1 month), and V2 (2-year follow-up ± 1 month) in PD patients. V1 vs V0, p<0.0001 (N=600); V2 vs V1, p<0.0001 (N=482); V2 vs V0, p<0.0001 (N=501); All groups; p<0.0001 (N=481). B. Change from V0 to V2 in the mean NMSS total score in PD patients (N=501) and controls (N=122). V2 vs V0 in controls (outlines in red), p=0.944; V2 vs V0 in PD patients (outlines in blue), p<0.0001; difference in the change from V0 to V2 between PD patients and controls, p=0.002. C. Mean score on each domain of the NMSS scale at V0, V1 and V2 in PD patients (blue) and controls (red).
Data (A and B) are presented as box plots, with the box representing the median and the two middle quartiles (25-75%). P values were computed using general linear models (GLM) repeated measures. Mild outliers (O) are data points that are more extreme than Q1 - 1.5 * IQR or Q3 + 1.5 * IQR”.
Regarding your suggestion to use raincloudplots we appreciate it but the representation of the data with conventional box diagrams is used in Figure 1 and we understand that it is correct.
As you suggested, in Figure 1C, we replaced the domain numbers by domain names. Thanks again.
Minor comments:
Introduction:
Please don’t refer to unpublished and non-accessible research articles (Ref 13 and its referring sentence)
AUTHORS – Thank you very much for your comment. This manuscript is under second review in a high impact factor journal (D1) and we hope that it could be published early. Abstract has been accepted for presentation in the next MDS 2021 Virtual Congress. The message about data observed in this cohort is important and is one of the justifications for this analysis and paper.
Methods:
It is not indicated whether the MedOFF or the MedON UPDRSIII was considered for the regression model for patients with motor fluctuations
AUTHORS – Thank you very much for your comment. UPDRS-III included in the models was during the OFF state. In tables (Table 1 and Table 2) appear as UPDRS-III (OFF). We add in the text in Methods, when we explain all variables included, the term “(OFF)”.
I suggest introducing the concept of mild, moderate, severe, or very severe NMS burden in the Methods section. Is there any reference for the chosen differentiation?
AUTHORS – Thank you very much for your comment. In Methods, the last paragraph before Data analysis explain this: “The global NMS burden was defined as the NMS total score and it was categorized in 4 groups: mild (NMSS 1-20); moderate (NMSS 21-40); severe (NMSS 41-70); very severe (NMSS > 70) [17]”. As it is indicated, the reference is the next: Ray Chaudhuri K, Rojo JM, Schapira AH, et al. A proposal for a comprehensive grading of Parkinson's disease severity combining motor and non-motor assessments: meeting an unmet need. PLoS One 2013;8:e57221.
Results:
Please indicate the disease duration at study inclusion
AUTHORS – Thank you very much for your comment. We included this information: “mean disease duration 5.5 ± 4.37 years”.
Discussion:
Do you have any thoughts on why the PDSS does not change in the PD cohort but change in PDSS is one predictor of NMS progression?
AUTHORS – Thank you very much for your comment. Although the mean value of a score of a scale may not vary globally, it does not mean that there could not be changes in the value of some patients that could have changes in the other comparative scale too and therefore, the increase in the PDSS total score could be related to the increase in the NMSS scale score. In fact, it was observed a moderate negative correlation between the changes from V0 to V2 in the PDSS and the NMSS total scores (r=-0.340; p<0.0001).
As stated by the authors, one of the limitations of the study is its early PD bias. I think it would be interesting to add a comparison of the results in this cohort to the results of a published advanced PD cohort (e.g. MED control group at 36 month follow up in Jost et al. 2020)
AUTHORS – Thank you very much for your comment. We included a comment about this in Discussion. In this study in PD patients from the MED group the NMSS total scale increased from (mean) 46 to 62.8 but it was not significant (p=0.072; N=38). The sample was small. We think that it is an excellent example of how NMS progression may be modified with an intervention. We add the sentence in Discussion: “As example, bilateral subthalamic stimulation (STN-DBS) improved significantly the global NMS burden and QoL in advanced PD patients after 36 months but not in patients who received the standard-of-care medical therapy [19]”.
Table 1:
Please indicate the effect size (e.g. cohen’s d) for each scale investigated in the PD group
AUTHORS – Thank you very much for your comment. As you suggested, we included this information in Table 1. In Methods with add the sentence: “Cohen´s d formula was applied for measuring the effect size. It was considered: small effect = 0.2; medium effect = 0.5; large Effect = 0.8”.
When the V1 data is included in the analysis (which is IMO not absolutely necessary), the results should also be reported in Table 1.
AUTHORS – Thank you very much for your comment. The analysis of this study includes the baseline evaluations (V0), at 2 years (V2), and the comparison between both evaluations considering patients and controls. The assessment at 1-year follow-up (V1) has only been considered as an intermediate point in patients to show the progressive change in the NMSS total score. It is described in Methods.
Figure 2
I suggest to indicate significant changes in the figure and not as text below it (A)
AUTHORS – Thank you very much for your comment. As you suggest, the change has been done.
I recommend a language editing of the manuscript as there are several typos, missing words, or results reported twice in one sentence, in the manuscript and the tables (also some Abbreviations are missing below the tables).
AUTHORS – Thank you very much for your comment. The manuscript was reviewed by a native English speaker from USA, Darrian McAfee, from the Univeristy of Pennsylvania. We think that there was a problem with the edition of the text (word).
Reviewer 2 Report
Dear Authors:
I carefully read this relevant study and I have only a few comments:
The term “PD patients” is currently not the most consensual, I suggest changing it to people or person with Parkinson.
I was at a loss to understand how the control group was recruited. I would like to have more information about this control group.
Author Response
Reviewer #2 (Comments to the Author):
I carefully read this relevant study and I have only a few comments:
The term “PD patients” is currently not the most consensual, I suggest changing it to people or person with Parkinson.
AUTHORS – Thank you very much for your comment. We agree with you that the term “people or person with Parkinson” can be useful for some texts, but in general, not scientific, more targeted to patients or caregivers (e.g., https://www.caregiver.org/resource/parkinsons-disease-caregiving/). In fact, as first author (DSG), I have experience in developing projects with associations of Parkinson's patients and they always insist on not using the term "patient" to avoid certain stigmatization. In the scientific literature, really is very frequent to use the term PD patients (e.g., https://bmcneurol.biomedcentral.com/articles/10.1186/s12883-019-1276-8; https://www.ncbi.nlm.nih.gov/pmc/articles/PMC7196180/; etc.). So, if there is not an inconvenience, we think that the term PD patients is correct in this case.
I was at a loss to understand how the control group was recruited. I would like to have more information about this control group.
AUTHORS – Thank you very much for your comment. Control subjects matched by age, sex and educational level were recruited. The control subject could be a family member (not the patient's caregiver) or friend of the patient who would like to participate voluntarily. In some of the centers, the controls recruited were health workers, that is, hospital workers who voluntarily agreed to participate. The same inclusion (except PD diagnosis) and exclusion criteria as those for the patients were applied.
Round 2
Reviewer 1 Report
Thank you for addressing this point regarding group comparisons. Nevertheless, the problem of multiple comparisons has also to be considered for multiple correlations unless you declare them as being conducted as post-hoc analysis.
Thank you or clarifying this. I understand that the aim of Fig 1A is to show all data you collected, but nevertheless it remains of limited value for me as it doesn’t show longitudinal progression of NMS in a PD cohort but cross sectional data in three different cohorts with a certain amount of overlap regarding the included individuals.
I also refer to my previous comment regarding Fig 1A. Regarding the visualizationand coloring of Fig 1B I still suggest keeping the same outline color and only changing the filling. But maybe the journal might have some suggestions for this.
By indicating I meant illustrating, not just adding the text inside the figure
Author Response
Dear reviewer,
Many thanks for your comments.
We have adedd the sentence in methods: The Bonferroni method was used as a post-hoc test after ANOVA.
We agree with you regarding Figure 1. Again, thanks for your comments for improving the manuscript. We have modified Figure 1A and only those patients with data at V0, V1 and V2 were included (N=481). The value at V0 changes from 45.52 ± 37.84 to 44.92 ± 37.87, at V1 from 47.61 ± 37.05 to 47.03 ± 35.88, and at V2 from 53.47 ± 42.28 to 53.32 ± 42.46 (p<0.0001 for all analysis). Moreover, in Figure 1A and 1B, the color of the outline and filling is the same righnow and it appears quite clear for understanding.
Finally, we provide a confidential versión of the manuscript under second review in other Journal (D1, FI > 6). If you need more information about it, please, ask for it.